# Pharmacological evidence for the implication of noradrenaline in effort

**Nicolas Borderies[1], Pauline Bornert[1], Sophie Gilardeau[2], Sebastien Bouret[1] ***

**1** Motivation, Brain and Behavior team, Institut du Cerveau et de la Moelle épinière (ICM), INSERM UMRS 1127, CNRS UMR 7225, Pitié-Salpêtrière Hospital, Paris, France, **2** Phenoparc PRIM'R, Institut du Cerveau et de la Moelle épinière (ICM), INSERM UMRS 1127, CNRS UMR 7225, Pitié-Salpêtrière Hospital, Paris, France

* sebastien.bouret@icm-institute.org

**Data Availability Statement:** All relevant data presented in the paper and in supporting material (includingt S1 Data) are available from the ICM Institutional website. In addition, data have been

## Abstract

The trade-off between effort and reward is one of the main determinants of behavior, and its alteration is at the heart of major disorders such as depression or Parkinson's disease. Monoaminergic neuromodulators are thought to play a key role in this trade-off, but their relative contribution remains unclear. Rhesus monkeys (*Macaca mulatta*) performed a choice task requiring a trade-off between the volume of fluid reward and the amount of force to be exerted on a grip. In line with a causal role of noradrenaline in effort, decreasing noradrenaline levels with systemic clonidine injections (0.01 mg/kg) decreased exerted force and enhanced the weight of upcoming force on choices, without any effect on reward sensitivity. Using computational modeling, we showed that a single variable ("effort") could capture the amount of resources necessary for action and control both choices (as a variable for decision) and force production (as a driving force). Critically, the multiple effects of noradrenaline manipulation on behavior could be captured by a specific modulation of this single variable. Thus, our data strongly support noradrenaline's implication in effort processing.

## Introduction

The balance between costs and benefits is crucial in many disciplines interested in behavior, and studies addressing the neurobiological processes underlying the arbitrage between effort costs and reward benefits provided a significant insight into this arbitrage [1–7]. Among the numerous structures potentially involved in the effort/reward trade-off, monoaminergic neuromodulatory systems play a central role. The 3 major monoaminergic neuromodulatory systems (dopaminergic, noradrenergic, and serotoninergic) are defined by the specific molecule that they are secreting, and their cell bodies are located in deep nuclei of the central nervous system [8–12]. These neuromodulatory systems are altered in several neurological and psychiatric diseases that include motivation disorders (major depressive disorders, schizophrenia, Parkinson's disease) [13–15]. Moreover, catecholamines are the target for the majority of pharmacological treatments for these diseases [16,17].

Among those neuromodulators, dopamine (DA) is thought to be particularly critical for motivation and the effort/reward trade-off [2,6,18,19]. Indeed, dopaminergic activity is reliably related to the value of the upcoming reward [20–22] and also to the vigor of the animal's

posted on the following public repository: https://
osf.io/5z7dq/.

**Funding:** Fondation de France (recherche en
psychiatrie) to SB. Recurrent founding from CNRS
and ICM to SB. The funders had no role in study
design, data collection and analysis, decision to
publish, or preparation of the manuscript.

**Competing interests:** The authors have declared
that no competing interests exist.

**Abbreviations:** DA, dopamine; GLM, general linear
model; LC, locus coeruleus; NA, noradrenaline; RT,
response time.

response [23–26]. Experimental manipulations of the dopaminergic system, using pharmacol-
ogy [27] or optogenetics [28], demonstrated the causal role of dopamine in the production of
motivated behaviors. But even if dopamine seems to be crucial for promoting resource mobili-
zation based on the upcoming reward, its relation to effort (when separated from reward)
remains debated. Effort generally refers to the amount of physical or mental energy needed to
reach a goal. In situations in which effort only refers to performance and scales with reward
through incentive motivation, measuring it as a specific process (distinct from reward and per-
formance) is very difficult. Intuitively, however, effort can readily be separated from reward
and incentive motivation, because we usually experience effort precisely when we need to per-
form a challenging task without the drive of an immediate reward. Based on this intuition, we
propose to study effort as a motivational process affecting choices and performance based on
the expected costs, independently of the incentive influence of rewards. Critically, our objec-
tive here is not to redefine the theoretical boundaries of effort. Rather, we aim at characterizing
a motivational process centered on expected costs, independently from the expected benefits,
both at a computational and at a neurobiological level.

In that frame, we refer to effort as a computational variable capturing the amount of
resources necessary for action, both when making a choice (and using it as a decision variable)
and when performing the action (and using it as a driving force for behavior). At the time of
the decision, one tends to minimize this variable and to choose the least effortful option. At
the time of the action, however, performance increases with the amount of mobilized resources
[7]. When using that definition of effort and when dissociating it from reward in experimental
paradigms, the relation between DA and effort became tenuous. For example, in recent experi-
ments measuring dopaminergic activity using either single-unit recordings or voltammetry, it
was much more sensitive to the amount of expected reward than to the level of effort [29–33].
Similarly, when effort, force, and reward could be dissociated, dopaminergic manipulation
had little effect on effort processing compared with its effect on reward processing, as mea-
sured using choices [34–37]. Again, we do not question the strong implication of dopamine in
energizing behavior and cognition [38,39], but here, our goal is to examine the possibility that
energy for action can rely upon a complementary motivational process, which is critical when
the action is not directly rewarded and which we will refer to as effort.

Several recent studies suggested that effort (as defined here) could rely upon the other
major catecholamine: noradrenaline (NA) [40–42]. The relation between NA and effort is
indeed supported by a several lines of evidences: First, locus coeruleus (LC) noradrenergic
neurons are systematically activated right before initiating a costly action [43–45], and the
magnitude of this activation relates to the action's level of difficulty. Indeed, at that time, the
firing of LC neurons scales positively with the costs and negatively with the reward levels, sug-
gesting that LC activation is related to the mobilization of resources necessary for overcoming
that difficulty [33,46,47]. In line with an interpretation in terms of physical effort, we recently
showed a strong effect of noradrenergic manipulation on the amount of exerted force in a task
in which force was not instrumental [48]. What remains unclear, however, is the extent to
which noradrenergic manipulations would affect effort, defined as a single computational vari-
able affecting both choices and force production. Indeed, in a previous work [48], noradrener-
gic manipulation did not affect choices when options were characterized by a trade-off
between reward size and reward-schedule length. This might be taken as evidence against the
role of NA in effort (assuming that schedule length involves effort), but because schedule
length has a strong delay component, it might affect decision through a pure delay discounting
process and therefore leave the relation between NA and effort open (see [7,49] for a discus-
sion on effort and delay). Second, pharmacological manipulations of the NA system strongly
affect performance in tasks requiring a high level of executive control [50–53] or attentional

set-shifting [54–56]. These data are directly in line with the key action of NA on executive functions and its neural substrate in the prefrontal cortex [57–59]. Intuitively, these data could be interpreted in terms of "mental effort," with the intuition that here effort would affect the trade-off between difficulty and performance in the cognitive rather than physical domain. Third, LC activity is closely associated with autonomic arousal and pupil dilation [33,60,61], which is itself correlated with both physical and mental effort [62–66]. Fourth, this relationship between LC/NA and effort also resonates with an extensive literature demonstrating the role of NA in vigilance and arousal, which constitutes a basic form of resource mobilization and appears related to effort [67–69]. But because we defined effort as a computational variable affecting both decisions and action execution, it cannot be captured by generic processes such as vigilance and arousal, which are less specific and can be readily separated from both physical and cognitive effort (for instance, arousal can increase with reward alone or surprise, even if no effort is involved whatsoever). In short, several lines of evidence converge to support the hypothesis that NA plays a key role in effort, but there is no direct evidence.

The goal of the present study is to address this issue directly by manipulating NA levels in monkeys performing a task involving the effort/reward trade-off. We designed a behavioral task that completely differentiated effort costs (a force to produce) from benefits (the expected reward magnitude). Because this task involved both binary choices and action execution (monkeys choose the option by executing it), we could reliably measure effort as a process involved both in choices and force production and separate it from confounding factors such as force itself, reward, or arousal. In order to establish a causal link between NA levels and effort, we used systemic pharmacological perturbations of noradrenergic transmission (clonidine). Because we hypothesized that NA was involved in effort, we expected clonidine to affect force production and to have a stronger effect on effort-based compared with reward-based choices. Moreover, we expected the 2 effects (choice and force) to be related, through a single computational variable, which we refer to as effort. For the most part, our results are compatible with this hypothesis, and they provide a strong support to the emerging idea that NA plays a central role in motivation. Thus, even if energy for action and cognition can be modulated by the expected benefits through dopamine-dependent incentive processes, this study provides strong evidence that the specific influence of the expected costs on resources mobilization and decision-making relies upon a NA-dependent process, which we refer to as effort.

## Methods

### Ethics statement

Experimental procedures were designed in association with veterinarians, approved by the Regional Ethical Committee for Animal Experiment (CREEA IDF no. 3), and performed in compliance with the European Community Council Directives (2010/63/UE) at the *Institut du Cerveau et de la Moelle Epinière* (Agreement number: A-75-13-19).

### Monkeys

Three rhesus monkeys (Monkey A, male, 8 kg, 9 years old; Monkey B, male, 15 kg, 8 years old; Monkey E, female, 6 kg, 7 years old) were used as subjects for the experiments. All monkeys have been properly habituated to the lab environment and used in another behavioral experiment before this one. Food was available ad libitum, and motivation was controlled by restricting access to fluid to experimental sessions, when water was delivered as a reward for performing the task. Animals received water supplementation whenever necessary (e.g., if they could not obtain enough water through work), and they had free access to water whenever testing was interrupted for more than a week.

## Experimental settings

**Drugs.** We selected a pharmacological agents that conformed to the following criteria: (1) selectivity for noradrenergic targets, (2) bibliographic references that could provide effective dose for the treatments, and (3) compatibility with human pharmacology for translational purposes [46,48,70]. We used clonidine, a selective α2-agonist known to decrease LC activity, with a fixed dose of 0.01 mg/kg. Solutions were freshly prepared at the beginning of the week by dissolution into a fixed volume of saline per animal. The control vehicle was a saline solution (NaCl 0.9%) matched in volume. Systemic intramuscular administrations were performed on the lateral side of the thigh. The pharmacological schedule followed a within-monkey, bloc-wise, multisession procedure. Animals received an injection every day, there were blocks of active treatment administration (3 to 5 consecutive injection within a week) intermixed with blocks of inactive vehicle after a minimal washout period of 48 hours [16]. Solutions were administrated at 30 minutes before testing for all sessions. For all monkeys, $n$ = 12 sessions were collected for active treatments and at least 12 sessions for control vehicle injections. Two monkeys (A, B) were tested simultaneously (on the same day) with similar drug conditions and another monkey (E) was tested independently (after completion of the full schedule for the 2 previous monkeys).

**Behavioral task.** Monkeys performed the behavioral task while squatting in a restraining chair, in front of a computer screen (HP Compaq LA2405wg 24 inches, LCD technology, 1,920 × 1,200 pixels at 60 Hz), with 2 homemade handgrips hooked on the chair. Handgrips were connected to pneumatic tubes converting the squeezing force of the animals into air pressure, which was transduced, amplified, and digitalized at 1 kHz. A calibration procedure ensured every day that the sensitivity of the system was stable over time. The maximal overshoot was 151% of the calibration force. Average empirical maximal forces were 102.61 ± 1.25% (mean ± SEM), meaning that calibrations were close to the empirical capacities of the monkeys. The monkeys were monitored in real time with a camera. Task execution and behavioral recordings were controlled by a single computer. The behavioral task was run under MATLAB 2013a (www.MathWorks.com) with Psychtoolbox-3.0.11 (www.psychtoolbox.org). Reward delivery (water) was controlled by a custom-made automatic pump.

The behavioral paradigm was a binary choice task in which monkeys chose between 2 options, each characterized by a trade-off between effort costs and reward benefits (*Fig 1*). The task was designed to evaluate the effort/reward trade-off both in terms of action selection (choice) and action execution (force production). In this task, monkeys had to choose between 2 options presented simultaneously on the left and right part of the screen by pressing the grip on the side of the chosen option. Each of the 2 options was composed of 2 attributes: reward size and required force level, which were manipulated independently and randomly across trials. Reward size corresponded to the amount of water delivered during the outcome period (4 levels; 1, 2, 3, or 4 drops of water), and required force level corresponded to the force threshold that had to be exceeded during the response period (20%, 40%, 60%, or 80% of maximal force). The experimental conditions differed between trials in their reward level (1, 2, 3, or 4 water drops) and required force (20%, 40%, 60%, or 80% of maximal force) assigned to each option (left or right). These conditions were assigned to a trial number with a random permutation of the 256 different conditions (4 reward levels and 4 required force levels for both left and right options), reinitialized at the end of each blocks of 256 trials and at the beginning of each new session (with an independent permutation). All monkeys had experience with tasks requiring exerting force on a grip before, but they were gradually trained to perform this task, as well as with the principle of choosing between 2 grips. Monkeys were trained until they reached asymptotic performance in this task, implying that they could reliably estimate the amount of required force necessary for each action based on the corresponding visual cue. We

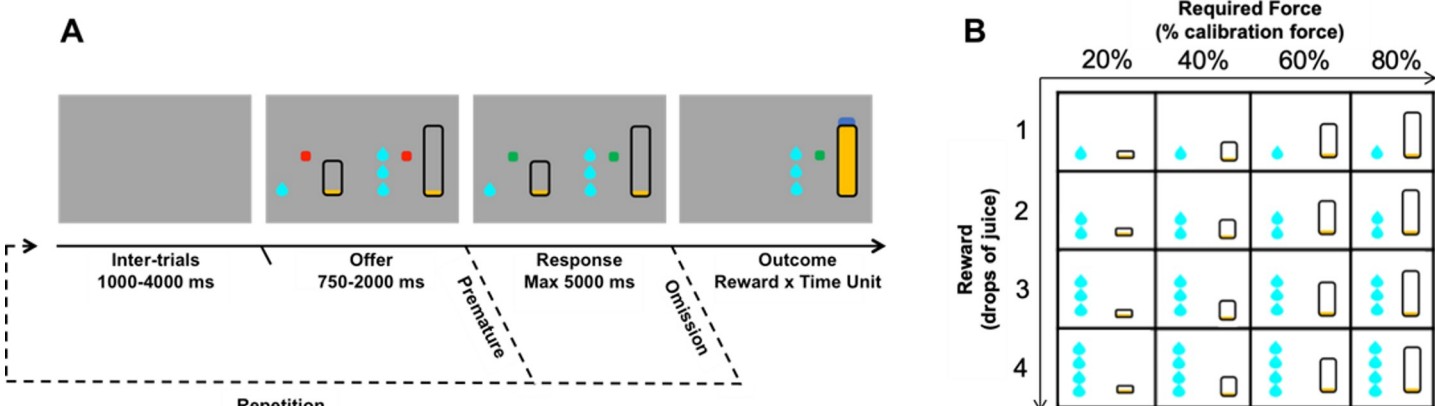

**Fig 1. Behavioral task design. A.** Schematic description of a trial sequence with successive screenshots displayed to the monkeys. A trial started with a gray screen (intertrial phase) followed by the presentation of 2 red instruction dots and 2 options, each of them composed of a reward cue and a required force cue (offer phase). When the instruction dots turned green (response phase), the monkeys could choose 1 option by pressing the corresponding handgrip (left or right). A continuous visual feedback was displayed to signal the online force exerted by the monkey (yellow filling of the required force cue). The announced reward (drops of juice) was delivered if and when the exerted force reached the target force (outcome phase). The same trial started again if the monkey responded before the end of the offer phase (premature error) or provided no response at all (omission). **B.** Experimental design for the combinations of required force and reward size used to define options. Each option, regardless of its side of presentation, was defined by a combination of a given reward level (1, 2, 3, or 4 drops) and a given required force (20%, 40%, 60%, or 80% of the calibration force). The effort and reward components of each option were manipulated systematically and randomly, such that the influence of effort and reward on choices were orthogonalized by design. All possible combinations of pairwise options were sampled during the task.

used these training sessions to estimate a maximal force level for each grip and for each monkey. Because this maximal force was used as a calibration reference in all experimental sessions (forces are expressed as a percentage of this maximal force), it was kept constant for the reminder of the study (this prevented 2 artefacts: variations of success rate due to between-sessions variations of difficulty or strategic participation of the monkeys due to between-sessions variations of difficulty).

Each trial started with the presentations of an offer (combination of 2 options), which lasted for a random interval (ranging from 750 to 2,000 milliseconds) during which the monkeys could inspect the options' attributes but not respond. In case of premature response during this period, the exact same offer was repeated at the next trial until a correct response was made. This ensured that monkeys carefully considered the 2 options before responding, thereby emphasizing the distinction between option evaluation and action. The beginning of the response period was cued by a visual stimulus (go signal, colored square), indicating that the monkey could select 1 of the 2 options by squeezing the handgrip corresponding to the chosen option on the side of visual presentation of the corresponding cue. In case of omission (when the monkey did not respond during the response period), the exact same options were repeated at the next trial until a choice was made. If the monkey appropriately exerted the level of force associated with the selected option on the corresponding handgrip, the reward associated to that option was delivered. Otherwise, the task simply skipped to the next trial. Trials were separated by a random interval of 1–4 seconds. Each session ended when the monkey stopped participating and responding to free reward delivery (approximately 45 minutes on average, across animals, which was sufficient to sample the whole range of options).

## Data analysis

**Behavior.** Effort was captured by assessing its effect on choices, (i.e., the weight of the costs associated to the required force level) and on action execution (i.e., the force exerted

during action execution, estimated using force peaks). Again, given the task design, the influence of effort could be studied independently from that of reward, both on choices and on force production. The number of trials per session was sufficient to allow for a complete and homogenous sampling of all combinations of required force and reward levels in these experiments. Note that effort costs and reward levels affected not only the selection between the 2 options of an offer but also the willingness to perform the task at all, which we assessed using participation frequency as a function of the offered options. The dataset contained 3 levels of observations that were treated differently: (1) trial-wise observations, (2) session-wise observations, and (3) monkey-wise observations. We summarized trial-wise observations with average scores (participation frequency, choice frequency, force peak) to assess aggregate effects, and with regression weights extracted from generalized linear models (logistic models for binomial data) to assess parametric effects. Only trials with an explicit choice expressed (no omission or anticipation) were selected in the analysis of choices and force peaks. For session-wise observations, all sessions were considered independent; therefore, we computed behavioral metrics separately for each session and monkey. Next, the session-wise and monkey-wise analyses were modeled together into a mixed model, the random effect of monkey identity was a covariate of no interest, and the fixed effects across sessions and monkeys were the effects of interest. Thereby, the experimental conditions of treatment type were modeled as mixed effects. Inferences on active treatment effect were conducted with parametric 2-sided, 2-sample $t$ tests onto fixed effect of active treatment versus vehicle.

All analyses were done with MATLAB 2017a software. General linear models (GLMs) and GLMM models were estimated with a maximum likelihood algorithm ("fitglm" and "fitglme" MATLAB functions). Computational model estimation was completed with an approximate maximum a posteriori algorithm. The procedure used a variational Bayesian technique under the Laplace assumption, implemented in a MATLAB toolbox (available at http://mbb-team. github.io/VBA- toolbox/).

### Computational modeling

**Multisource-source modeling of the modulation of behavior by effort cost/reward benefit trade-offs.** Lastly, we used a mathematical model derived from decision-theory principles to explain the variety of behavioral markers of interest in a unified framework (**S1 Fig**, **S1 Table**). The model assumes that the behavior of the monkeys resulted from an effort cost–reward benefit stochastic optimization, similarly to previous work [35]. The exact implementation of the theoretical model for the present behavioral task had 2 main purposes: (1) integrating the various recorded behaviors into a unifying framework (enabling us to get a session-by-session estimate of the cognitive construct of interest, namely effort) and (2) disentangling the direct influence of effort cost–reward benefit valuations on behavior from potential confounding processes (e.g., behavioral excitation [here captured with $K_0$] or subjective uncertainty [or risk, here captured by $\sigma_E$]).

This model can be decomposed into 2 main components, valuation and selection, respectively captured in Eqs 1–4 and 5–7. The value of each option is computed by subtracting potential costs from potential benefits, based on the attributes of each options (Eq 1): the reward benefit term (modulated by the reward level, R), an effort cost term (a quadratic function of the force level F, see [71–73] for similar formulation), and a subjective probability of success conditional to the exerted force (see [74] for similar formulation), when the purpose of the valuation is to select the appropriate force, given the information about required force (Eoption) (Eq 4). Option value is computed with the assumption that the optimal force $F^*$ is applied (Eq 2), meaning that the best scenario is considered at the valuation stage. The

influence of rewards and efforts decreases over time because of satiety and fatigue, respectively, which are captured by the corresponding variables and the number of trials in the session (N). Each of these terms are parametrized by constants controlling the behavioral predictions given by the model; those constants can be divided into 4 categories: (1) sensitivities to action-outcome values ($K_{R:}$ reward sensitivity, $K_{E:}$ effort sensitivity); (2) behavioral response tendencies ($K_{0:}$ activation weight, $\sigma_{E:}$ action-outcome uncertainty); (3) dynamic sensitivities ($K_{S:}$ satiety sensitivity, $K_{F:}$ fatigue sensitivity, $K_{N:}$ activation-dynamic sensitivity) and (4) behavioral distraction tendencies ($b_{M:}$ motor bias, $b_{P:}$ perseveration bias). The second equation (Eq 2) captures 3 types of behavior: participation (whether monkeys are willing to engage in the task or not, at every trial), the choice itself (which of the 2 options is selected), and the exerted force (how much force the animal actually produces during action execution). Choices are predicted by a softmax decision function integrating the different option values, whereas the exerted force is the one optimizing the value of the chosen option. Complex functions in these equations are specified in Eqs 8, 9 and 10.

**Valuation equations**

Force valuation:

$$V(F|option) = \kappa_0(1 + \kappa_N N)F + \kappa_R(1 + \kappa_S N)P(R_{option}|F)R - \kappa_E(1 + \kappa_F N)F^2 \tag{1}$$

Option valuation:

$$V(option) = V(F^*|option) \tag{2}$$

Offer valuation:

$$V(active) = \sum_{i=left,right} V(option_i) \tag{3}$$

Subjective success rate evaluation:

$$P(R_{option}|F) = 1 - normcdf(E_{option}|F, \sigma_E) \tag{4}$$

**Selection equations**

Participation:

$$P(participation = 1|options_t) = logistic(V(active) + b_P P(participation = 1|options_{t-1})) \tag{5}$$

Choice:

$$\begin{aligned} &P(choice = right|options_t, participation_t = 1) \\ &= logistic(V(option_{right}) - V(option_{left}) + b_P P(choice = right|options_{t-1}) + b_M) \end{aligned} \tag{6}$$

Force:

$$F = \arg\max_F(V(F|options_t)) + \varepsilon(\sigma_F) \tag{7}$$

**Definitions**

$$\varepsilon = \mathrm{N}(0, \sigma_{\mathrm{F}}) \tag{8}$$

$$logistic(x) = \frac{1}{1 + e^{-x}} \tag{9}$$

$$normcdf(x|\mu, \sigma) = \frac{1}{\sigma\sqrt{2\pi}} \int_{-\infty}^{x} e^{-(x-\mu)^2/2\sigma^2} dx \tag{10}$$

The model captures the influence of the cost/benefit trade-off onto participation (Eq 5) and choice (Eq 6). These 2 behaviors are the results of a soft-max selection between the different options: Participation rate was determined by the offer value (the value of the 2 options), and choices were determined by the value difference between the right and left option. Intuitively, the reward-benefit term increases the rate of participation (for options with a maximal impact of the marginal benefit of participation on value) and biases choices towards more rewarding and effortful options (amplifying the effect of reward difference while blunting the effect of effort difference). The effort-cost term has an opposite influence, decreasing participation for all conditions (without main dependence on the amount of expected rewards and required force), and biasing choices towards less rewarding and effortful options (blunting the effect of the reward difference while amplifying the effect of effort difference). The activation component simply increases participation in a similar pattern as the reward benefit component and biases choices towards less effortful options (because it automatically amplifies the global force anticipated for all levels of required force). The uncertainty component decreases participation by diluting the effect of the benefit component (therefore counteracting its impact for low-reward high-effort options) but does not induce significant switch of the choice rate (because the model already assumes that the adaptation to reduce uncertainty of demanding effort occurs during the adjustment of exerted force).

One of the core features of this model is the selection of the exerted force, i.e., how much force the animal will exert on the grip given the expected costs and benefits (Eq 7). The 2 components of value (reward and effort) act in opposite directions: the reward-benefit term (scaled by reward-benefit sensitivity) enhances the exerted force, whereas the effort cost term (scaled by required force-cost sensitivity) decreases the exerted force in a nonlinear (quadratic) fashion [35]. Note that because the force cannot take negative values, it goes directly to zero when the expected value of the action is negative (i.e., when action is less valuable than inaction). As a common practice [35], we included a noise term ($\varepsilon$), which is scaled by the standard deviation of the residuals ($\sigma_{\mathrm{F}}$). Obviously, the 2 components of value will control (in opposite directions) the adaptation of force to experimental conditions (expected reward and required force) such that the reward benefit term enhances the exerted force for low rewards (in which the marginal benefit of force has maximal impact on value) and the effort-cost term decreases the exerted force for low rewards and high required forces (in which the marginal cost of force has maximal impact on value). The activation component (scaled by the activation weight, $K_0$) acts as a benefit term independent from the expected reward, increasing exerted force and blunting the effect of experimental conditions (this would lead to a maximal force exerted for all rewards and effort combinations at saturation). The uncertainty component ($\sigma_E$) does not have a clear effect on the direction of force modulation (because it depends on the dual impact of uncertainty on the marginal value of force and on the net value of force). By contrast, uncertainty has a straightforward effect on reward and its influence on behavior, including force production. The marginal expected success is directly weighted by the reward term in the

equation. Note that the nuisance terms were not included in the exerted force equation for the sake of simplicity. In summary, the dynamic sensitivities to reward benefits and effort costs control the allocation of force over time, in interaction with the value components for satiety and fatigue sensitivities.

**Bayesian model estimation.** The different models were inverted using a variational Bayesian approach under the Laplace approximation implemented in a MATLAB toolbox (available at http://mbb-team.github.io/VBA-toolbox/; [75]). This algorithm not only inverts nonlinear models with an efficient and robust parameter estimation (through a maximum a posteriori procedure) but also estimates the model posterior probability, which represents a trade-off between accuracy (goodness of fit) and complexity (degrees of freedom).

### Reporting

The main text contains only major statistical results (group-level effects for variable of interest); however, exhaustive reporting of the statistical results can be found in **S2–S7 Tables**.

## Results

### Behavioral task

We designed a 2-options reward-force choice task to evaluate the behavioral markers of effort-costs/reward-benefits arbitrages both in terms of choice and force production (*Fig 1*). At each trial, the monkey was required to make a choice between 2 options presented on the left and right part of the screen. Each of these options was characterized by 2 attributes: reward size and required force. Reward size corresponded to the volume of water delivered in correct trials, and trials were considered as correct when monkeys produced a force that exceeded the threshold defined by the required force (in terms of percentage of the maximal force, as defined in the Methods). As depicted in *Fig 1*, options' attributes (force and reward) were visually cued: Reward size was indicated by a number of drops and required force by the height of a rectangle. We refer to the offer as the pair of options presented at a given trial. The presentation of the offer lasted for a random interval (ranging from 750 to 2,000 milliseconds) during which the monkey could inspect the options' attributes. Then, a visual go signal indicated the beginning of the response initiation period, during which the monkey could choose 1 of the 2 options by squeezing the corresponding handgrip. Finally, if the monkey successfully exerted the required force, i.e., exceeded the force threshold for the selected option, the corresponding reward was delivered. In case of failure to exert the required force, the task simply skipped to the next trial, which was selected randomly. In case of omission (no squeezing either grip), the exact same offer was repeated at the next trial until a choice was made. Participation (squeezing any grip, successfully or not) and choices (selection of 1 of the 2 options by pressing on the corresponding side) were treated separately. The experimental conditions differed between trials in their reward level (1, 2, 3, or 4 water drops) and required force (20%, 40%, 60%, and 80% of maximal force), randomly and independently assigned to each option (left or right) (*Fig 1*). Only sessions that included more than 100 trials were included in the analysis. This corresponded to the average number of trials required to explore once each combination of reward and force options without taking into consideration the side of presentation $[(4 + 3 + 2 + 1)^2]$. The analysis included 148 sessions (48 for monkey A, 52 for monkey B, 48 for monkey E) with a median length of $n = 361$ trials (431 for monkey A, 278 for monkey B, 408 for monkey E) and an interquartile range of 219 trials (212 for monkey A, 104 for monkey B, 225 for monkey E).

## The task mobilizes an effort/reward trade-off

We first assessed the validity of the task by evaluating the impact of task parameters on monkeys' behavior. We tested whether reward size and required force affected force production, participation, and choices using generalized linear models (*Fig 2*).

First, the force exerted by the monkeys was a reliable indicator of the monkey's compliance to the task rule (*Fig 2A*): force peak increased systematically with the required force of the chosen option ($\beta = 0.06$, $t_{(147)} = 4.45$, $p = 1.69 \times 10^{-5}$), demonstrating that monkeys understood the relationship between visual cues and the amount of force to exert on the grip. Note that force exertion was also affected by the chosen reward ($\beta = 0.02$, $t_{(147)} = 3.25$, $p = 1.42 \times 10^{-3}$) (*Fig 2B*), and the interaction between reward and required force was significant ($\beta = 0.01$, $t_{(147)} = 2.97$, $p = 3.50 \times 10^{-3}$).

Participation reflects the monkeys' willingness to perform the task and select one of the 2 options, given the offer. In the analysis of participation, we asked 2 questions: (1) whether offer attributes (required forces and rewards) influenced trial-by-trial participation and (2) whether this influence reflected an effort cost/reward benefit trade-off. To answer the first question, we imagined 2 scenarios for the influence of offer attributes on participation:

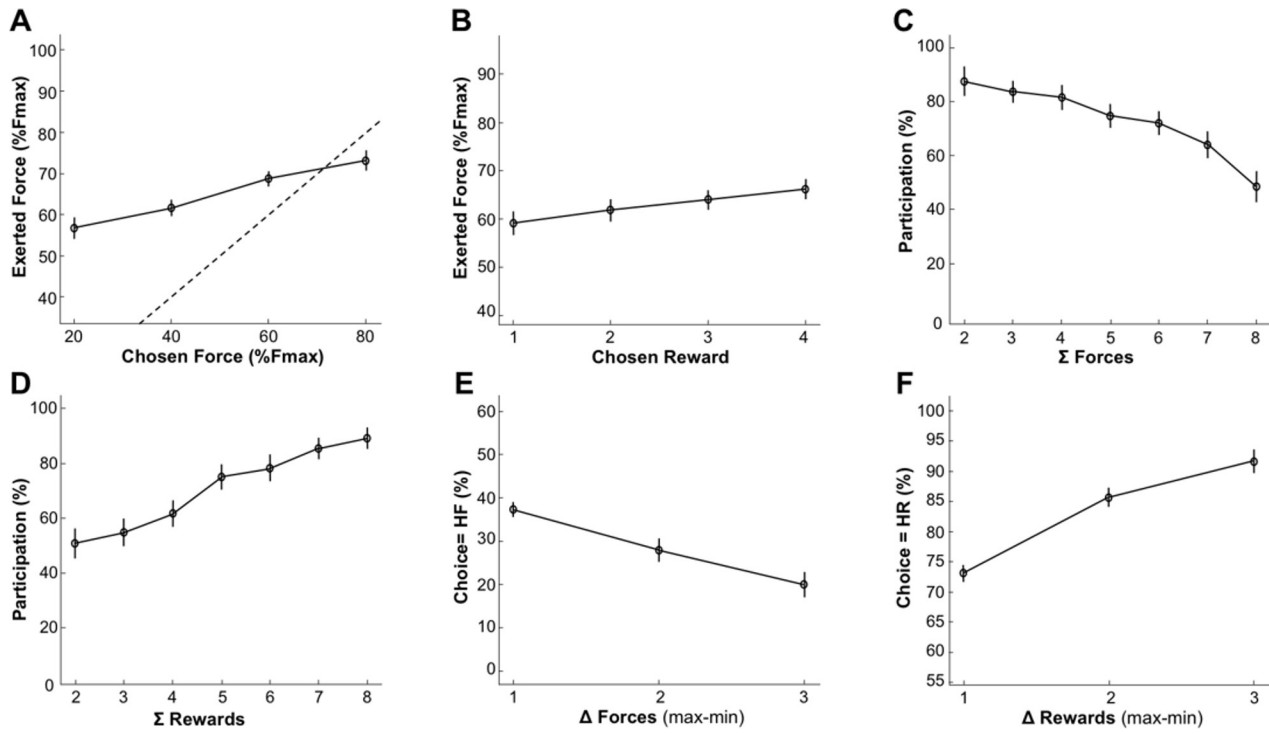

**Fig 2. Behavior under placebo condition.** Lines and dots represent group-level means (across monkeys), and error-bars represent SEM (across sessions). **A**. Relationship of exerted force with required force of the chosen option. The force exerted at the peak increased with the required force of the chosen option ($\beta = 0.06$, $t_{(147)} = 4.45$, $p = 1.69 \times 10^{-5}$). The dotted line represents the optimal relationship between exerted force and required force (the minimum required force to complete the trial, i.e., the position of the force threshold). **B**. Relationship between exerted force and expected reward of the chosen option. Reward had a positive influence on the exerted force ($\beta = 0.02$, $t_{(147)} = 3.25$, $p = 1.42 \times 10^{-3}$). **C-D.** Participation (%) to trials as a function of the sum of required forces (**C**) or reward (**D**). **C**. Participation was negatively modulated by the sum of required forces ($\beta = -0.51$, $t_{(147)} = -7.93$, $p = 4.99 \times 10^{-13}$) presented in an offer. **D**. Monkeys' participation was positively modulated by the sum of rewards ($\beta = 0.49$, $t_{(147)} = 6.62$, $p = 7.36 \times 10^{-5}$) presented in an offer. **E**. Proportion of high-reward choices as a function of the difference in offered rewards between the 2 options (1D-reward conditions). There was a significant effect of the difference in required force on choices ($\beta = -1.84$, $t_{(147)} = -6.95$, $p = 1.09 \times 10^{-10}$). **F**. Proportion of high-required-force choices across difference in offered forces (1d-required force conditions). There was a significant contribution of the difference in reward in choices ($\beta = 2.39$, $t_{(147)} = 9.44$, $p = 7.94 \times 10^{-17}$). Underlying data can be found in S1 Data. Fmax: maximal force; HF, high force; HR, high reward.

(Scenario 1) Monkeys adjusted their participation based on the value of only 1 option in the offer—the best option, which is the optimal strategy—or (Scenario 2) participation was influenced by the relative value of the current offer (defined by the rewards and required forces of the 2 options) on the scale of what could be expected in this task, in terms of reward sizes and required force levels. In that frame, the relative value of each offer would simply scale with the sum of the value of the 2 options (sum of rewards–sum of forces). We tested these 2 hypotheses by comparing a model predicting participation based on the largest reward and the smallest required force with a model in which participation depended on the sum of rewards and the sum of required forces. Compared with the latter, the former model was very unlikely (*freq* = 0.37, *ep* = 3.6×10$^{-3}$). Hence, monkey's participation more likely arose from an evaluation process involving both options (Scenario 2). To address the second question, i.e., whether this process reflected an effort-cost/reward-benefit trade-off, we selected the best model (sum of option values) to test for a consistent effect of reward and required force attributes onto the participation. This confirmed that participation was positively modulated by the sum of rewards offered ($\beta$ = 0.49, $t_{(147)}$ = 6.62, $p$ = 7.36×10$^{-5}$) (**Fig 2D**) and negatively modulated by the sum of offered required forces ($\beta$ = −0.51, $t_{(147)}$ = −7.93, $p$ = 4.99×10$^{-13}$) (**Fig 2C**), in line with an effort cost/reward benefit trade-off.

We also examined the monkeys' choices, which appeared reliably dependent upon the difference between option values (**Fig 2E and 2F**). There was indeed a significant contribution of the difference in both reward ($\beta$ = 2.39, $t_{(147)}$ = 9.44, $p$ = 7.94×10$^{-17}$) (**Fig 2F**) and required force ($\beta$ = −1.84, $t_{(147)}$ = −6.95, $p$ = 1.09×10$^{-10}$) (**Fig 2E**). Even if the previous analyses clearly demonstrates that monkeys adjusted their behavior based upon visual information about expected reward and required force, suggesting an effort-reward trade-off, the true nature of the cost as a decision variable influencing choices negatively might not have been effort but risk. Here, risk is defined as the probability of failing to complete a trial, knowing that the monkey participated. Indeed, monkeys tended to fail more often in trials with high required force (logistic regression of the success rate, $\beta$ = −3.24, $t_{(147)}$ = −3.98, $p$ = 1.05×10$^{-4}$), such that monkeys could have avoided options of high required force just because they were associated with higher risks (of not obtaining the reward) rather than with higher effort. To address this issue, we performed a model comparison between models predicting behavior using required force or success rate as predictor variables. Models integrating the required force level fit our data better than the other ones integrating risk for both participation (*freq* = 0.58, *ep* = 9.96×10$^{-1}$) and choices (*freq* = 0.98, *ep* = 1). Thus, behavior in this task provides reliable estimates of reward/effort trade-offs in monkeys.

Finally, we used a computational approach to capture the joint influence of the effort-costs/reward-benefits trade-off on choices and force production (see **Fig 3**, Methods; details of the models can be found in **S1 Fig** and **S1 Table**). In short, effort is defined as a single computational variable having a dual influence on behavior: a negative influence on choices (all other things being equal, monkeys minimize energy expenditure) and a positive influence on force production (monkeys exert more force and therefore spend more energy for high-force options). The model assumes that monkeys tried to optimize the ratio between effort-costs (the amount of resources needed to perform the task, modulated by fatigue accumulating during the task) and reward benefits (the expected reward, given the force produced, as well as the level of satiety that builds up over time. When compared with the behavioral data, the model exhibited a good fit of participation (balanced accuracy = 72%), choice (balanced accuracy = 73%) and peak force ($R^2$ = 16%). It was also able to remarkably reproduce the qualitative effect of experimental conditions (**Fig 4**). This computational approach confirms the interpretation of the behavioral data in terms of effort, defined as the amount of resources invested in the action and affecting both choices (as a variable of decision) and force production (as a

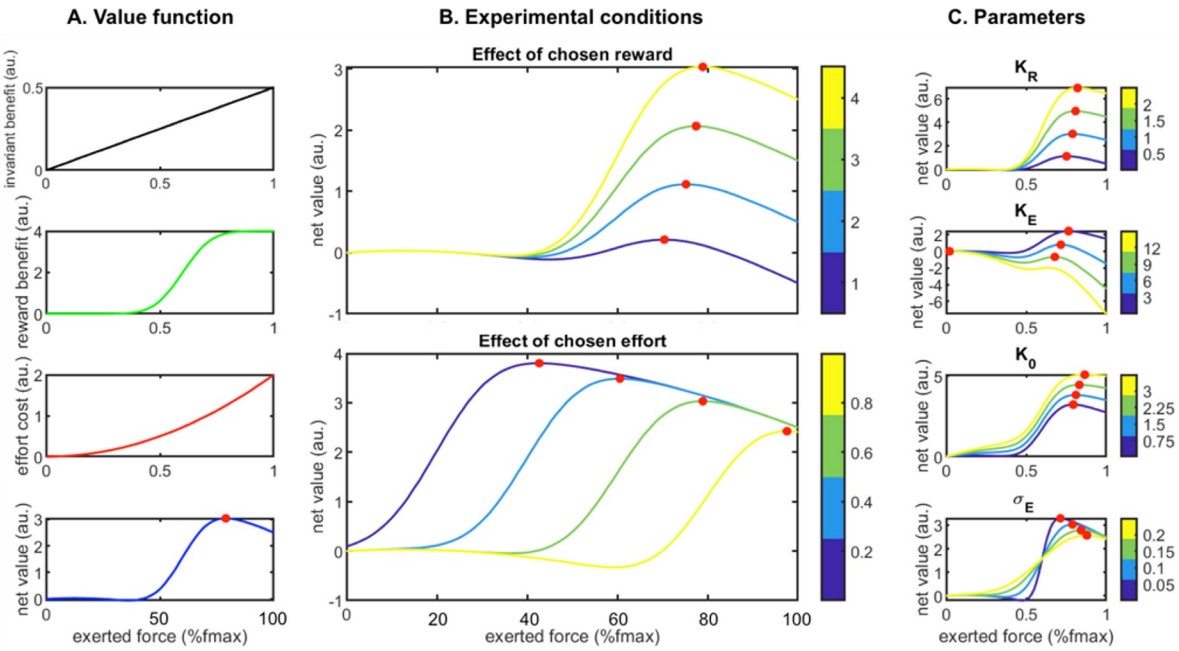

**Fig 3. Properties of the optimal force selection model. A**. Components of the value function depending on the exerted force. **A1**: the activation component is a linear invariant benefit, **A2**: the reward benefit component is a sigmoidal function with a slope scaled by the uncertainty component, **A3**: the effort cost component is a quadratic function. **A4**: The net value is the sum of 3 previously described components. **B**. Impact of main experimental conditions onto the value function: **B1**: chosen rewards increase the optimal expected value and slightly shift the optimal force. **B2**: chosen effort decreases the optimal expected value but increases the optimal force (bottom) **C**. Impact of main parameters onto the value function. **C1**: increase of $K_r$ (sensitivity to reward) leads to huge increase of optimal value and slight increase of optimal force. **C2**: increase of $K_e$ (sensitivity to effort) leads to decrease of optimal value and decrease of optimal force with a collapse to zero-force when the optimal value becomes negative. **C3**: increase of $K_o$ (activation weight) leads to increase of optimal value and increase of optimal force. **C4**: increase of $\sigma_E$: (*action-outcome uncertainty*) leads to decrease of optimal value and increase of optimal force, shifting it away from the target force (bottom). Simulations were performed with an initial parameters vector of {= 1; = −1; = 2; = 1; = 0.5; = −1; = 0.1; = 0; = 0; = 0} and an initial input vector of {R = 4; E = 0.6; N = 0}. Red dots indicate the optimal force to exert when the net value function is represented. Color-bars indicate either input or parameter values when there are variations. au., arbitrary units; Fmax: maximal force.

driving force). Altogether, this analysis of the behavioral data indicates that monkeys did follow an effort/reward trade-off in this task, with effort negatively influencing the choice and positively influencing action execution, and reward size positively modulating both of these processes.

## Clonidine increases effort sensitivity without affecting reward sensitivity

To assess the role of NA in the effort/reward trade-off, we examined the influence of clonidine, a drug that decreases NA levels, on the previously described behavioral measures ([46,70]) (***Fig 5***). First, clonidine had a significant influence on choices, and the nature of that influence was particularly obvious in some task situations. For example, in offers for which 1 of the 2 options included both the highest level of force and the highest level of reward (referred to as the high-reward/high-required-force option), the frequency of choosing that high-reward high-required-force option was decreased (*clonidine = 0.47, placebo = 0.63*, $F_{(1,145)} = 43.54$, $p = 7.28 \times 10^{-10}$), indicating a shift in sensitivity to expected effort and/or to the expected reward. To evaluate the specific influence of the drug on reward-based versus effort-based choices, we selected conditions in which options differed only in reward levels (1D-reward conditions) and conditions in which options differed only in required force (1D-required force conditions). Practically, we computed the frequency of high-reward choices in the 1D-reward

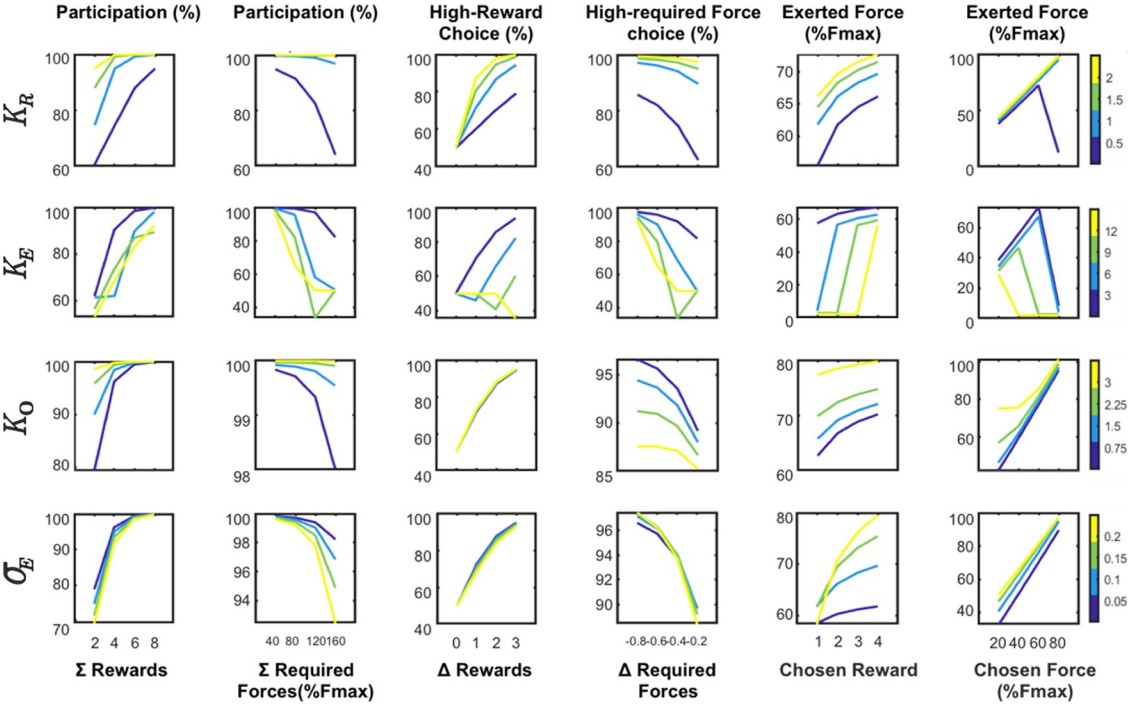

**Fig 4. Predictions of the complete computational model for choice and force production.** The graphs display major predicted dependencies between experimental conditions and measurements under changing parameter values. The 6 columns display the relationship between **1.** the participation rate and sum of offered rewards. **2.** The participation rate and sum of offered efforts. **3.** The proportion of high-reward choices and difference in offered rewards. **4.** The proportion of high-effort choices and difference in offered required forces. **5.** The exerted force and the chosen reward. 6. The chosen effort. The rows display the impact of the 4 key parameters (top to bottom: $K_r$, sensitivity to reward; $K_e$, sensitivity to effort; $K_o$, activation weight; $\sigma_E$, action-outcome uncertainty.). Color bars indicate parameter values. Simulation*s were performed with an initial parameters vector of {= 1; = −1; = 2; = 1; = 0.5; = −1; = 0.1; = 0; = 0; = 0} and an initial input vector of {R = 4; E = 0.6; N = 0}*. Fmax: maximal force.

conditions and the frequency of high-required-force choices in the 1D-required force conditions (**Fig 5A and 5B**). There was a specific decrease in the proportion of high-required-force choices in the 1D-required force conditions (*clonidine = 0.14, placebo = 0.24*, $F_{(1,145)}$ = 14.81, $p$ = 1.78×10$^{-4}$) (**Fig 5A**) but no significant effect on high-reward choices in the 1D-reward conditions (*clonidine = 0.86, placebo = 0.86*, $F_{(1,145)}$ = 9.5×10$^{-3}$, $p$ = 0.92) (**Fig 5B**). Finally, to evaluate the relative influence of reward and effort information on choices across all task conditions, we used a logistic regression to model choices in every single trial of the task, for both placebo and clonidine sessions. This analysis confirmed that the weight assigned to the difference in required force levels between options increased under clonidine (*clonidine = −2.55 ± 0.23 [mean and SEM], placebo = −1.65 ± 0.28*, $F_{(1,145)}$ = 14.81, $p$ = 1.78×10$^{-4}$), whereas the drug did not affect the weight assigned to the difference in reward sizes (*clonidine = 2.29 ± 0.21, placebo = 2.45 ± 0.30*, $F_{(1,145)}$ = 0.47, $p$ = 0.49).

After assessing the impact of clonidine on choices, we considered its influence on participation as a function of the required forces and reward sizes of the offers (sets of 2 conditions, **Fig 5C and 5D**) using a mixed model in which monkey identity was treated as random effects and all other factors as fixed effects. Paradoxically, monkeys participated more under clonidine treatment (*clonidine = 0.68, placebo = 0.62*, $F_{(1,145)}$ = 5.82, $p$ = 1.71×10$^{-2}$), a phenomenon characterized by a significant increase in participation bias (the constant term of the logistic regression; *clonidine = 1.33 ± 0.30, placebo = 0.79 ± 0.26*, $F_{(1,145)}$ = 5.01, $p$ = 0.02). Besides of this increase in participation bias, there was also a tendency for an increase in the negative weight

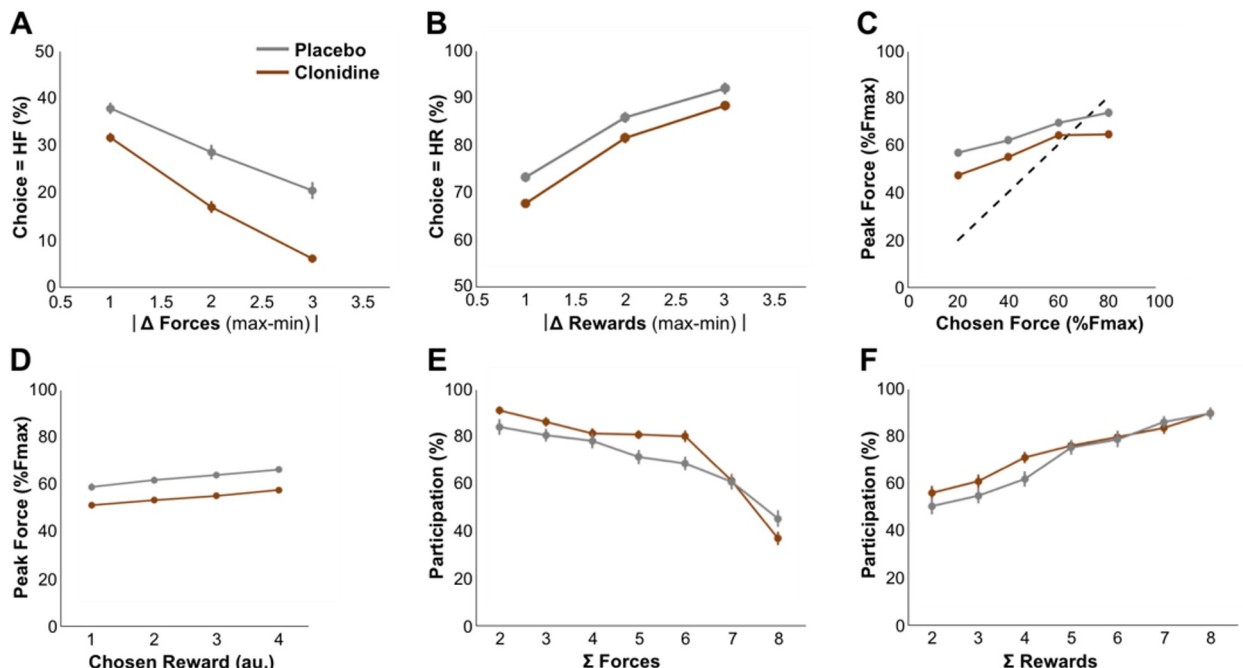

**Fig 5. Clonidine affects choice and force production.** Behavior of the monkeys under the placebo (gray) and clonidine (brown) conditions. Lines and dots represent group-level means (across monkeys) and error-bars represent SEM (across sessions). **A.** Relationship between the percentage of high-required force choices and the difference in offered forces. The weight of the difference in required force increased under clonidine (clonidine = $-2.55 \pm 0.23$, placebo = $-1.65 \pm 0.28$, $F_{(1,145)} = 14.81$, $p = 1.78 \times 10^{-4}$). **B.** Relationship between the percentage of high-reward choices and the difference in offered rewards. Under clonidine, the weight of the difference in reward sizes did not change (clonidine = $2.29 \pm 0.21$, placebo = $2.45 \pm 0.30$, $F_{(1,145)} = 0.47$, $p = 0.49$). **C-D.** Relationship between exerted force relation and chosen force level (**C**) and chosen reward size (**D**). The dotted line represents the optimal relation (identity) between exerted force and force difficulty (the minimum required force to complete the trial). Clonidine had a main effect on force production (clonidine = $0.55 \pm 0.05$, placebo = $0.63 \pm 0.03$, $F_{(1,145)} = 20.70$, $p = 1.12 \times 10^{-5}$), without affecting the effects task factors (reward weights: clonidine = $0.02 \pm 8.3 \times 10^{-3}$, placebo = $0.02 \pm 5 \times 10^{-3}$, $F_{(1,145)} = 0.93$, $p = 0.33$; required force weight: clonidine = $0.06 \pm 0.01$, placebo = $0.05 \pm 0.01$, $F_{(1,145)} = 0.92$, $p = 0.33$). **E-F.** Relationship between participation and the sum of offered forces (**E**) and sum of offered rewards (**F**). There was a tendency for an increase in the weight of the sum of required forces that did not reach significance (clonidine = $-0.75 \pm 0.18$, placebo = $-0.46 \pm 0.04$, $F_{(1,145)} = 2.02$, $p = 0.16$) (**E**) but no effect of clonidine on the modulation of participation by the sum of rewards (**F**). Underlying data can be found in S1 Data. au.,arbitrary units; Fmax, maximal force; HF, high force; HR, high reward.

of the sum of required forces that did not reach significance in the fixed-effect component (*clonidine* = $-0.75 \pm 0.18$, *placebo* = $-0.46 \pm 0.04$, $F_{(1,145)} = 2.02$, $p = 0.16$) but was significant for 2 out of 3 monkeys (random effects, Monkey A, $F_{(1,145)} = 37.46$, $p = 8.27 \times 10^{-9}$; Monkey B, $F_{(1,145)} = 15.92$, $p = 1.04 \times 10^{-4}$; Monkey E, $F_{(1,145)} = 2.05$, $p = 0.15$). The positive effect of clonidine on the participation bias might be a side effect of the increased sensitivity to effort that reduces the average total reward per trial and thus might induce a compensatory increase in participation to maintain the total amount of reward per session. Consistent with this explanation, there was no difference in the total reward earned between clonidine and placebo sessions (*clonidine* = $269.99 \pm 32.41$ ml, *placebo* = $272.11 \pm 35.25$ ml, $F_{(1,145)} = 6.4 \times 10^{-3}$, $p = 0.93$), despite the decrease in high required force/high-reward choices (see previous). Furthermore, participation biases across clonidine sessions were negatively predicted by session-wise choice sensitivity to required force ($\beta = -0.51 \pm 0.08$, $t_{(145)} = -6.15$, $p = 7.14 \times 10^{-9}$), supporting an interpretation in which the influence of clonidine on a single latent decision variable (effort costs) had both a negative influence on required force-dependent choices and a positive influence on participation, through a compensatory mechanism to ensure sufficient water intake.

Clonidine also affected force production negatively, as measured by a significant decrease in the maximum exerted force (force peak) after clonidine compared with placebo injections

(*clonidine* = 0.55 ± 0.04, *placebo* = 0.64 ± 0.03, $F_{(1,145)}$ = 21.78, *p* = 6.91×10$^{-6}$). The maximal force recorded within experimental sessions were significantly reduced under clonidine compared with placebo sessions (empirical Fmax(placebo) = 94.42 ± 3.86%; empirical Fmax(clonidine) = 105.75 ± 1.91%; F(1,145) = 10.4, *p* = 1.55e-03). The GLM performed on force peaks to decompose the influence of multiple factors showed that the drug only had a negative effect on the constant term (the positive bias to exert force on the grip, *clonidine* = 0.55 ± 0.05, *placebo* = 0.63 ± 0.03, $F_{(1,145)}$ = 20.70, *p* = 1.12×10$^{-5}$), but it did not affect the modulation of exerted force by either task factors (required force weight: *clonidine* = 0.06 ± 0.01, *placebo* = 0.05 ± 0.01, $F_{(1,145)}$ = 0.92, *p* = 0.33, **Fig 5E,** reward weights: *clonidine* = 0.02 ± 8.3×10$^{-3}$, *placebo* = 0.02 ± 5×10$^{-3}$, $F_{(1,145)}$ = 0.93, *p* = 0.33, **Fig 5F**). Thus, clonidine biased force production negatively, but it did affect its modulation by task parameters (reward and required force).

Altogether, the effect of clonidine onto choices and force production could be subsumed under a common increase of the sensitivity to effort cost. As an empirical confirmation of this intuition, there was a positive correlation between the parameters capturing each of these 2 effects across clonidine sessions. The parameter *βe* capturing the influence of required force level on choices and the intercept parameter (*βo*, bias) in the model capturing the modulation of force production displayed a significant positive relationship (*β* = 0.63± 0.10, $t_{(144)}$ = 6.48, *p* = 1.33×10$^{-9}$). Thus, across treatment sessions, there was a systematic positive relationship between the influence of clonidine on required force-based choices and on force production, in line with the idea that clonidine acts on a single variable, effort, which affects these 2 behavioral measures.

To confirm that interpretation, we used the computational model to capture the influence of the treatment on the effort variable. The first step was to check that the model provided a good fit for all 3 key variables: participation (balanced accuracy = 76%), choice (balanced accuracy = 76%), and peak force ($R^2$ = 21%) for clonidine sessions. Given the quality of the fit, we were justified in extracting the 10 parameters calculated on a session-by-session basis. We started by testing whether the set of parameters belonging to the placebo conditions were different from the clonidine session with a multivariate analysis of variance (predicting the parameter set with the treatment condition), enabling us to estimate the number of relevant dimensions separating the 2 treatment conditions. Only 1 dimension was sufficient to separate the 2 treatment conditions (d = 1, ë = 0.69, *p* = 6.37×10$^{-5}$), which means that a single hyperplane in the parameter space can provide a reasonable classification of treatment conditions. To further characterize the effect of clonidine on the computational parameters, we conducted a classification analysis (predicting the treatment condition with the parameter set) under sparsity assumption (a lower number of predictive parameters is more likely) to infer whether a specific parameter of the model was modified under the clonidine condition (and therefore could be a good predictor of the clonidine condition in the classification). We could reliably predict the treatment condition based on the computational parameters (balanced accuracy = 72%, excedance probability = 1 − 4.59×10$^{-4}$). Further details on the effect of the treatment on computational metrics can be found in **S7 Table**. We found 3 nonzero parameters that predicted the clonidine condition: an increase of the force cost sensitivity $K_E$ (+1.14 SD), an increase of the laterality bias $b_M$ (right-oriented) (+0.76 SD), and a decrease of the perseveration bias $b_P$ (−0.69 SD). The modification of the side bias, a behavioral nuisance parameter, could be due to the physical implementation of the required force (reflecting the fact that monkeys perceived the right handgrip as less difficult and thus selected it more often, all others things being equal). But more importantly, the force sensitivity that drives the 3 behavioral measures (participation, choice, and force production), was the single parameter of interest being impacted, and there was no effect on reward sensitivity, behavioral activation weight, or subjective uncertainty. In other words, the computational analysis confirmed that clonidine

influenced a single variable, "effort," affects both choices and force production as a function of expected costs but independently of expected benefits.

Altogether, this analysis demonstrates that clonidine administration increased the sensitivity to effort costs, i.e., increases the negative effect of required force on choices and decreased the amount of force produced. To further assess the specificity of this effect of clonidine on motivational processes, we considered more generic effects on behavior. First, we considered a global negative effect on behavioral reactivity (related to arousal or vigilance), rather than a specific effect on effort. As mentioned above, however, participation rate was increased under clonidine and response time was unaffected (*clonidine* = 0.53 ± 0.06, *placebo* = 0.52 ± 0.08, $F_{(1,145)}$ = 0.10, *p* = 0.75). Thus, the influence of clonidine on behavior in this task could not be captured solely as a global decrease in behavioral reactivity. We also examined the influence of clonidine on response times (RT). A simple regression analysis of RT could not retrieve any significant effect of clonidine, neither at the beginning of the session (RT(placebo) = 0.505 ± 0.087 seconds; RT(clonidine) = 0.539 ± 0.069 seconds; F(1,145) = 1.366, *p* = 0.244) nor on linear variations across the session (ΔRT/Δtime(placebo) = 0.045± 0.042 seconds; ΔRT/Δtime (clonidine) = −0.011 ± 0.031 seconds; F(1,145) = 1.385, *p* = 0.241). This does not exclude the possibility that clonidine affected the vigilance of monkeys because the trend is toward a slowing down of monkey responses. However, it could totally be a motivational effect that impact decision uncertainty, a major factor in the determination of response time (see **S1 Text** for a detail description of RT analysis). Second, we considered a direct effect of clonidine on muscle effectors, rather than on a decision variable computed in the brain. We measured the efficiency of muscular contraction through the known linear relationship between maximal force contraction and maximal velocity of contraction, also called Fitts law (see ref [35] for more details). The muscular contractility index (i.e., the regression coefficient between the velocity peak and the force peak) was not affected by clonidine treatment (*clonidine* = 4.32 ± 0.43, *placebo* = 4.14 ± 0.40, $F_{(1,145)}$ = 1.19, *p* = 0.27) (**Fig 6A**). Thus, clonidine did not impact the monkeys' capacity to execute the actions, and its influence on behavior was more likely due to an impact on motivation, more precisely, on the effort processing. Finally, we considered a scenario in which clonidine only had a direct effect of performance, but an indirect effect on choices (via the resulting decrease in reward probability). Under such an "indirect" scenario, clonidine only affected force production and consequently decreased the success rate of monkeys, making options requiring higher forces appear more risky (higher probability of failing). In other words, the influence of clonidine on choices would have been mediated by its effect on success rate. However, we observed no progressive change of the high-required-force choices during the course of clonidine sessions, as would be expected with a progressive adaptation to a decrease in success rate ($\beta$ = −9.73×10⁻⁵ ± 9.13×10⁻⁵, $t_{(145)}$ = −1.06, *p* = 0.28) (**Fig 6B**). The effect of clonidine on force was stable across trials (no main effect of trial number on peak force, **Fig 6C**). Furthermore, we did not observe any significant relationship between the success rate and the proportion of high-required-force choices across sessions ($\beta$ = 0.04 ± 0.06, $t_{(144)}$ = 0.70, *p* = 0.17) (**Fig 6D**). Finally, we directly examined the possibility that success rate was a better predictor of choices than required force by adding it as a covariate in the logistic regression model describing the influence of task parameters on choices, as well as the effect of clonidine. This did not affect the model fit, indicating that success rate had no significant role in mediating the influence of clonidine on required force-based choices ($F_{(1,145)}$ = 18.63, *p* = 2.93×10⁻⁵). In sum, the influence of clonidine on behavior is unlikely to result from a global decrease in behavioral reactivity, as expected for an interpretation in terms of vigilance or arousal. It is also unlikely to result from a direct effect on muscular effectors, and an indirect influence on choices though success rate. Rather, in line with our

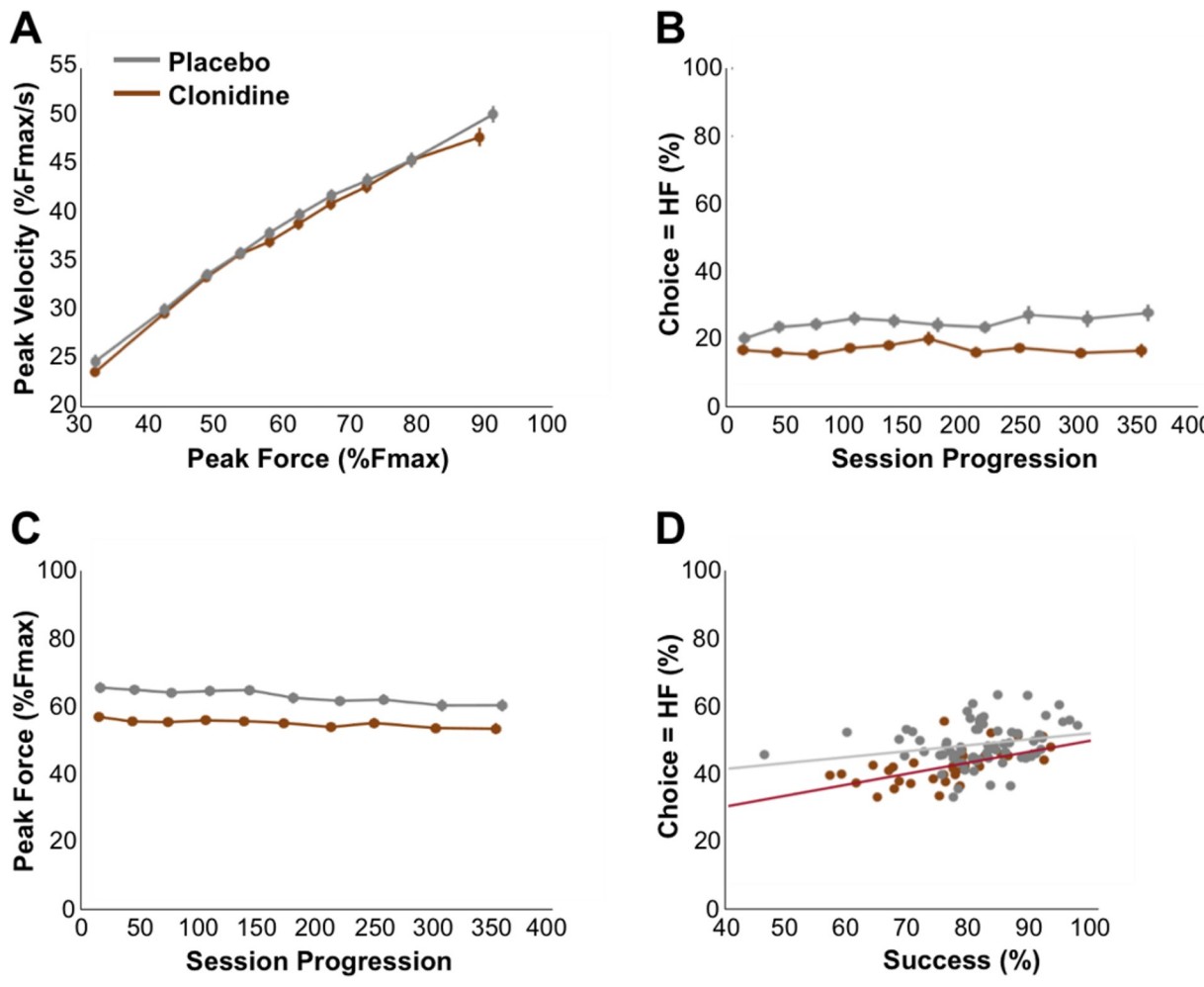

**Fig 6. Specificity of the clonidine effect.** Behavior of the monkeys in the placebo (gray) and clonidine (brown) conditions. Lines and dots represent group-level means (across monkeys) and error-bars represent SEM (across sessions). **A**. Relation between peak velocity and peak of the exerted force (commonly known as Fitts law). Clonidine did not affect this proxy for muscular contractility (clonidine = 4.32 ± 0.43, placebo = 4.14 ± 0.40, $F_{(1,145)}$ = 1.19, $p$ = 0.27). **B.** Influence of session progression (trial number) on the percentage of high-required-force choices. Monkeys made less high-required-force choices overall, but the effect was constant over the session ($\beta$ = −9.73×10$^{-5}$ ± 9.13×10$^{-5}$, $t_{(145)}$ = −1.06, $p$ = 0.28). **C.** Influence of session progression (trial number) on exerted force. Exerted force was decreased under clonidine, but the effect of the drug was constant over the session ($p$ > 0.05). **D.** Proportion of high-required force choices as a function of success (proportion of correctly executed actions in the session, across sessions). There was no significant relationship between success rate and the proportion of high-force choices ($\beta$ = 0.04 ± 0.06, $t_{(144)}$ = 0.70, $p$ = 0.17). Underlying data can be found in S1 Data. Fmax Maximal force,; HF, High force.

hypothesis, clonidine affects a single decision variable, which we refer to as "effort" and which affects both choices and force production.

## Discussion

### Summary

In the present study, we investigated the causal implication of NA in the effort/reward trade-off in monkeys performing a novel task in which (1) the influence of effort and reward could be dissociated and (2) this trade-off affected 2 distinct motivational processes: choice and force production. We examined the causal role of NA using systemic injections of clonidine, which decreased brain NA levels, and we used saline injections in control sessions. Clonidine

amplified effort costs, as measured both by a reduced fraction of high-force choices and by a reduction in the amount of force produced. This effect was relatively specific: Clonidine did not affect reward-based choices, and its effect could not be interpreted in terms of a global decrease in behavioral reactivity or in terms of motor impairment. Altogether, this work strongly supports the role of NA in effort, which we defined as a computational process adjusting the amount of resources mobilized for action. In that frame, the NA-dependent variable (« effort ») affects not only choices, as resource expenditure tends to be avoided, but also action production, as this increase in resources at the time of action drives performance.

## Technical considerations

This novel behavioral task enabled us to clarify the role of NA in effort by eliminating confounding factors. Here, we defined effort as a computational process reflecting the dual influence of resource mobilization on choices and force production. We believe that it is critical to clarify the definition of effort and avoid confusion with other variables such as mere exerted force, risk, or reward, in order to assess the relative contribution of distinct neuromodulatory systems to the distinct components of decision-making [7,9,10]. First and foremost, by independently manipulating reward and required force, we could isolate the direct influence of expected effort costs (required force) from the incentive influence of reward on force production. We clearly do not imply that this is a natural situation or that these experiments were meant to capture any ecological phenomenon. Clearly, animals must balance energetics costs and benefits to survive in their natural environment, such that ultimately effort and reward are intimately related [76]. In humans, effort and reward can become complex and multifaceted theoretical constructs and display nontrivial interactions that remain difficult to capture in other animals [77–79]. But again, our goal here was to evaluate the causal relation between NA and effort (defined here in a very specific way), using an artificial task in which it could be dissociated from reward. We admit that these are limitations.

Although monkeys had unequal performances across force levels, we verified that the effects of required force on behavior were more likely to be accounted for by effort than by risk (probability of failing) or muscular efficacy. Even if choices were more robustly predicted by the level of difficulty than by the probability of obtaining the reward after selecting an option, we cannot exclude that animals used the amount of risk associated with each difficulty level to decide which option to choose. Along the same lines, the amount of force produced by the monkeys (vigor) decreased over time, which was probably related to satiety and/or fatigue. Even though our task was not designed to study fatigue, we cannot exclude an interaction between fatigue and effort [80], but this could be addressed in future studies.

We used a systemic pharmacological approach to modulate noradrenergic levels in the brain and assess the causal role of NA in effort processing. Indeed, our goal here was to evaluate the functional role of NA through its global impact on the brain, without restricting our study to one of the numerous targets of the LC. Even though the functional heterogeneity of the noradrenergic system has been emphasized in recent studies, our study aims at characterizing the noradrenergic system as a single functional entity [81–83]. Given its widespread impact on global brain functions, we believe that a global manipulation of NA levels provides critical information regarding its functional role [84–87]. Another limitation of such pharmacological modifications is our limited knowledge of the actual impact of those treatments onto the noradrenergic system (LC firing rate, synaptic cleft NA concentration, post-synaptic potentials). Here, we take for granted that our observations result from the primary effects of the drug, as it remains the most parsimonious explanation for any observed pharmacological effect, and we therefore neglect the secondary effects. An important body of literature has demonstrated

that clonidine leads to decreased LC activity ([46,60,70]). Still, regarding its secondary effects, we cannot exclude the possibility that clonidine has a potential impact onto the hetero-inhibitory $\alpha_2$ receptors located in the prefrontal cortex [69]. However, the stimulation of prefrontal $\alpha_2$ receptors was shown to facilitate persistent activity in the prefrontal cortex and working memory performances [87,88], which is hard to reconcile with our findings regarding the effect of clonidine. In spite of these limitations, we hope that this study, by providing a clear insight into the role of NA in decision-making, will inspire future studies to dissect its underlying neurobiological mechanisms.

## Dual impact of clonidine on effort-based choices and force production

Clonidine had a strong and specific effect on effort processing, as monkeys tended to avoid options requiring more force and globally, they exerted less force. These effects are unlikely to be accounted for by a global decrease in reactivity or by a motor impairment: Animals participated more in the task under clonidine, and our measure of motor functions displayed no significant effect of the treatment. We could also exclude an interpretation in terms of risk (through a motor impairment) because both choices and participation were better explained by the level of required force than by success rate. Noticeably, the strong effect of clonidine on effort-based choices in this task contrasts with the lack of effect of clonidine on cost sensitivity in a task in which monkeys chose between sequences of different lengths to obtain their reward, and cost was interpreted in terms of delay rather than effort [48]. Altogether, our work shows that clonidine specifically affects the processing of effort costs without affecting decisions based on reward availability. This provides neurobiological support to the idea that effort differs from other types of costs such as delay or risk, which both relate to the distribution of reward in time and rely more critically upon dopamine [7,20,22,33,72,89,90]. More generally speaking, this work points to a very complementary role of NA and DA in effort, defined more generally as a mobilization of resources for action: The NA system would be involved in mobilizing energy based on expected costs (what, here, were referred to as "effort"), whereas DA would be more critical in mobilizing energy based on expected benefits (reward).

Besides affecting effort-based decision-making, clonidine also decreased the amount of force produced, irrespectively of the instruction. This confirms and extends previous observations in which clonidine decreased the amount of force produced in a task in which force was not instrumental [48]. Interestingly, in this task, the influence of clonidine on force production across sessions was directly related to its influence on decision-making, in line with the idea that it affects a common process, effort, which affects both choices and action execution. Critically, computational modeling confirmed that such a computational variable could reliably capture the monkeys' behavior in control conditions as well as the influence of clonidine. Altogether, these data are compatible with the idea that by decreasing NA levels, clonidine disrupts the monkeys' ability to mobilize resources to face the challenge at hand.

## Relation with LC neurophysiology

This interpretation is in line with our recent neurophysiological studies demonstrating a strong relationship between the activation of LC neurons and the triggering of demanding actions. Indeed, it is now firmly established that a transient LC activation occurs when animals trigger a goal-directed behavioral response [43–46]. Quantitative analyses of this activity revealed that the magnitude of the LC activation associated with the triggering of an action increased when the action value decreased, because its magnitude increased with lower reward and/or higher cost [33, 46, 47]. Note that the LC activation related to the difficulty to trigger the action (just before action onset) could readily be dissociated from the activation related to

the execution of the effortful action itself [33]. In sum, the fact that effortful events are associated with a transient activation of LC neurons is in line with the present findings suggesting that a minimum NA level is necessary to cope with effort. To some extent, this relationship between LC activation and effort is related to an emerging literature showing a strong link between cognitive load and pupil dilation, because pupil is sometimes interpreted as a proxy for central noradrenergic tone [62–66,91,92]. Interestingly, Zenon and colleagues already proposed to use pupil dilation as a physiological measure of effort to overcome either physical or mental challenges [42]. Note, however, that even if the relationship between pupil diameter and LC activity has been demonstrated both at rest and in relation to task performance [33, 61], it is far from being specific. Indeed, pupil dilation is essentially a measure of autonomic arousal, and as such, it is associated with numerous other autonomic and neuronal responses [61,93]. Thus, even if physiological, imaging, and pharmacological data coherently support an interpretation of NA function in terms of effort, the physiological processes underlying this function require further investigation, in particular to understand how the activation of the central noradrenergic system interacts with the rest of the brain to support behavioral and cognitive processes underlying effort.

### NA and effort: Beyond the physical domain?

Even if this study focused on physical effort, other studies have demonstrated the strong implication of NA in various forms of attention or executive control, which could be interpreted in terms of cognitive effort [50–53,55,56,94]. Even though this would require further testing, we believe that the physical and mental effort are actually 2 facets of the same core "effort" function in which NA plays a crucial role: To some extent, the influence of systemic manipulations of NA levels could all be interpreted in terms of allocation of cognitive resources. Indeed, even if clonidine affected physical effort, it affected the control of behavior (how much force the animals were willing to exert) rather than directly changing the reactivity of muscular effectors. In other words, our data are compatible with a theoretical frame in which NA levels define the amount of cognitive control available to overcome the challenges at hand, irrespectively of the nature of the challenge and the resources to be mobilized. Again, this is very speculative, and assessing the extent to which the role of NA in effort could be understood in terms of cognitive control would require extensive work, both at the computational and at the experimental level.

### Clinical relevance

These findings may provide an important neurobiological addition to recent frameworks for pathological deficits in motivation such as apathy, observed in neurological disorders such as Parkinson's disease and psychiatric disorders such as depression [3, 14, 17, 38]. Indeed, a deficit in motivation could be related both to an alteration of the incentive system (which enhances energy based on expected benefits, and relies upon dopamine) and an alteration of an "effort" system that enhances energy based on expected costs (as we propose here to rely upon NA). Being able to evaluate the relative impact of the disease upon each of these systems and/or to assess the efficacy of complementary strategies based upon DA versus NA may, in the long run, be beneficial to patients suffering from apathy.

## Conclusion

In sum, the current study demonstrates the causal role of NA in effort, defined as a process that mobilizes resources in order to face challenges at hand. We do not claim that this work provides a definitive description of effort, neither at a theoretical not at a neurobiological level. But because this interpretation crystallizes several elements of the literature on NA and

complements the huge literature on the incentive effect of dopamine, we hope that it will pave the way for further research investigating the physiological mechanisms by which the LC interacts with other brain regions to support effort.

## Supporting information

**S1 Fig. Graphical structure of the computational model.**
(TIFF)

**S2 Fig. Distribution of experimental conditions across placebo and drug sessions for all monkeys.** Absolute numbers of trials are indicated inside each box. Note that there is a difference between the placebo and drug session in the absolute numbers (because there were more placebo sessions included) but no difference in the relative proportions. *Fig 1*. Distribution of experimental conditions across placebo and drug sessions for all monkeys. Absolute numbers of trials are indicated inside each box. Note that there is a difference between the placebo and drug session in the absolute numbers (because there were more placebo sessions included) but no difference in the relative proportions. Underlying data can be found in S1 Data. dE, difference of force levels between the 2 options; dR, difference of reward levels between the 2 options.
(TIFF)

**S3 Fig.** Exerted force depicted as a function of the chosen reward (in abscises) and the required force level (with different line styles), for the placebo (in gray) and the clonidine sessions (in brown).
(TIFF)

**S1 Data. Processed data used to construct the visualization of** *Fig 2*, *Fig 5*, *Fig 6* **and** S3 Fig.
(XLSX)

**S1 Text. Analysis of response time in the reward-effort choice task.**
(DOCX)

**S1 Table. Description of the inputs, outputs, and parameters of the computational model.**
(XLSX)

**S2 Table. Summary of treatments impact on main behavioral metrics.**
(XLSX)

**S3 Table. Summary of treatments impact on ethological metric.**
(XLSX)

**S4 Table. Summary of the logistic regression coefficients for participation rate.**
(XLSX)

**S5 Table. Summary of the logistic regression coefficients for choice rate.**
(XLSX)

**S6 Table. Summary of the regression coefficients for peak force.**
(XLSX)

**S7 Table. Summary of treatments impact on computational metrics.**
(XLSX)

## Acknowledgments

The authors would like to thank Caroline Jahn for providing critical preliminary data and valuable input regarding the experimental design and the project as a whole. We would like to

thank Julia Mattioni for assistance with manuscript preparation. We also would like to thank the ICM Phenoparc Core Facility, especially Estelle Chavret-Reculon and Morgane Weissenburger for continuous assistance ranging from animal care to precious advices regarding these experiments.

## Author Contributions

**Conceptualization:** Nicolas Borderies, Sebastien Bouret.

**Data curation:** Sebastien Bouret.

**Formal analysis:** Nicolas Borderies, Sebastien Bouret.

**Funding acquisition:** Sebastien Bouret.

**Investigation:** Nicolas Borderies, Sophie Gilardeau, Sebastien Bouret.

**Methodology:** Nicolas Borderies, Sophie Gilardeau, Sebastien Bouret.

**Project administration:** Sebastien Bouret.

**Resources:** Sebastien Bouret.

**Software:** Sebastien Bouret.

**Supervision:** Sebastien Bouret.

**Validation:** Sebastien Bouret.

**Visualization:** Sebastien Bouret.

**Writing – original draft:** Nicolas Borderies, Sebastien Bouret.

**Writing – review & editing:** Nicolas Borderies, Pauline Bornert, Sebastien Bouret.

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
