## [Editor Report · Decision Letter 0]

20 Apr 2020

Dear Dr Bouret, 

Thank you for submitting your revised manuscript entitled "Pharmacological evidence for the implication of noradrenaline in effort" for consideration as a Research Article by PLOS Biology.

Your new version has now been evaluated by the PLOS Biology editorial staff, as well as by the original Academic Editor. I am writing to let you know that we would like to send your submission out for external peer review. As mentioned in our previous decision letter, we will invite a set of new reviewers, possibly overlapping with the past cohort.

Before we can send your manuscript to reviewers, we need you to complete your submission by providing the metadata that is required for full assessment. To this end, please login to Editorial Manager where you will find the paper in the 'Submissions Needing Revisions' folder on your homepage. Please click 'Revise Submission' from the Action Links and complete all additional questions in the submission questionnaire.

Please re-submit your manuscript within two working days, i.e. by Apr 22 2020 11:59PM.

Kind regards,

Gabriel Gasque, Ph.D.,

Senior Editor

PLOS Biology

---

## [Decision Letter · Decision Letter 1]

2 Jun 2020

Dear Sebastien,

Thank you very much for submitting a revised version of your manuscript "Pharmacological evidence for the implication of noradrenaline in effort" for consideration as a Research Article at PLOS Biology. This revised version of your manuscript has been evaluated by the PLOS Biology editors, by the original Academic Editor, and by two of the original reviewers: reviewers 1 and 3. In addition, as mentioned previously, we also invited a fourth independent reviewer to assess this new version. 

In light of the reviews (below), we are pleased to offer you the opportunity to address the comments from reviewer 4 in a revised version that we anticipate should not take you very long. We will then assess your revised manuscript and your responses, and we may consult this reviewer again.

We expect to receive your revised manuscript within two weeks. 

2. Together with the clean copy of the manuscript, please also upload a 'track-changes' version of your manuscript that specifies the edits made. This should be uploaded as a "Related" file type. 

In addition to the remaining revisions and before we will be able to formally accept your manuscript and consider it "in press", we also need to ensure that your article conforms to our guidelines. A member of our team will be in touch shortly with a set of requests. As we can't proceed until these requirements are met, your swift response will help prevent delays to publication.

*Copyediting*

*Published Peer Review History*

*Early Version*

*Submitting Your Revision*

Sincerely,

Gabriel Gasque, Ph.D., 

Senior Editor

PLOS Biology

DATA POLICY:

We note that you wrote in your Data Availability Statement: “Data are available from the ICM Institutional Data Access for researchers who meet the criteria for access to confidential data.” While we honor legal and ethical restriction to data sharing, these usually apply to human data. Could you please explain with these data are confidential?

Note, however, that we do not require all raw data. Rather, we ask that for the individual quantitative observations that underlie the data summarized in the figures and results of your paper. These data can be made available in one of the following forms:

Regardless of the method selected, please ensure that you provide the individual numerical values that underlie the summary data displayed in the following figure panels: Figures 2A-F, 5A-F, 6A-D, and S2. 

Please also ensure that the figure legends in your manuscript include information on where the underlying data can be found and ensure your supplemental data file/s has a legend.

Reviewer remarks:

Reviewer #1: The revised manuscript addresses all concerns raised in the initial review.

Reviewer #3: The authors have answered very completely the critiques of the three reviews. In particular, I was pleased with the clarifying passages in the new version of the manuscript. This is an important paper that separates effort and reward in a fashion that should have been standard in this area, but only recently has drawn more serious attention. I commend the authors on their thoughtful and response to review and I have no further questions or critiques.

Reviewer #4: Borderies et al., present a neuropharmacology study in monkeys, aimed assessing the causal role of noradrenaline in effort-based decisions and the exertion of force. Using a physical effort-based task and the pharmacological agent clonidine, they show that reducing noradrenaline levels reduced the willingness to choose to exert effort, and exertion of force. 

Overall, this an is interesting study, that tests a well-justified hypothesis, that noradrenaline levels are linked to effort processing. Although the results are not particularly surprising or a major advance given the labs existing data (Varazzani et al., 2015, JoN), and at times the manuscript could be much more clearly written, it is neat to have a causal demonstration of a link between noradrenaline and effort. Overall, the authors have done a good job of addressing the existing reviewer's comments and I will not add significantly to the burden of the authors by making many additional points. However, I do have some comments that I think warrant being addressed. 

Major

1. In the revisions the authors argue that the results could not be explained by risk, despite the subjects showing an undershoot in force exerted at the higher levels. However, I am not entirely convinced by the response, as from my reading it cannot be determined whether there is a contribution of risk to the effort discounting process in this work. I think that this should be explicitely stated as a limitation in the Discussion, no matter whether the authors believe it unlikely. 

2. I like the evidence presented that effort based choice and effort production could be unified. But, I wondered how much this is down the paradigm, compared to how often it is that effort and reward become linked together. This was a point raised by other reviewers, that I did not feel was adequately addressed. The reason I believe it is important as there is lots of evidence that at a more abstract level, effort and reward can become contingent on each other (Pooresmaili et al., 2015, PNAS), effort can sometimes become valued (Inzlicht et al., 2019, TICS) and sometimes there are contexts when choosing to exert effort does not map on to being willing to exert as much force - such as in the social condition of Lockwood et al., 2017, Nature Human Behaviour. Again, I think the authors should more simply state in the discussion that this experiment is an example where effort and reward can be dissociated, but such a dissociation depends on the context, with many situations where effort and reward become linked or choice and force production become unlinked, which may complicate the simple story of noradrenaline increasing motivation to exert effort. 

3. Several of the reviewers noted that the definition of effort was not clear. The authors have attempted to clarify, this in paragraph 2 and 3 of the introduction. However, I still found the definition hard to follow. There are several grammatical mistakes in the paragraph making it more challenging. Would they not be better off saying that there is (i) The objective difficulty of a task, (ii) the degree to which a subject is willing to choose to exert the effort required to complete that level of task difficulty and (iii) the energisation of the action, and thus exertion of effort, to complete the task. Following that the aim of this study was to design an experiment that dissociated effort and reward in both decision-making and force production. This would seem simpler than the current lengthy description and definition in those paragraphs. 

4. Changes across trials - A previous reviewer raised the concern about changes in vigour over trials. Although the authors suggest there is a trend, but it is not a major concern, I do think it would be better to say this explicitlyis a limitation and could be addressed in future studies in the discussion. There are of course studies showing that fatigue impacts on the willingness to exert effort for reward (Meyniel et al., (2013); Meyniel et al., 2016; reviewed in Muller & Apps, 2019). I don't think the task is set up to examine these effects appropriately, and this should be noted as a limitation in the discussion. 

5. In the abstract it refers to a "single hidden variable" as one of the key results. However, it is not until the manuscript has been read that it is clear what this actually means. This makes understanding the abstract very difficult. The authors need to clarify this result in the abstract much more clearly. 

MINOR

6. The manuscript state that the model was similar to the quadratic model of klein-Flugge et al. (2015) and Hartmann et al., (2013). However, the model in Klen-Flugge et al., was in fact a sigmoidal model, whereas in Hartmann et al., (2013) the model was parabolic. The authors should be more clear in the description here of whether it is a sigmoisal or parabomic model, as has been used more extensively in the literature (see Chong et al., 2017, PLoS Bio). 

7. It would help the ease of understanding of the paper, if the if Kr, Ke etc were defined in every figure legend where they are included. For instance, in figure 4 they aren't defined, which adds to the burden for the reader. 

8. There are many grammatical errors throughout the manuscript. As there are no line numbers it became difficult to note these down. However, I would suggest a native speaker is given the manuscript to read through, to correct the many errors in the text.

---

## [Editor Report · Decision Letter 2]

2 Sep 2020

Dear Dr Bouret,

On behalf of my colleagues and the Academic Editor, Christopher Summerfield, I am pleased to inform you that we will be delighted to publish your Research Article in PLOS Biology. 

Early Version

PRESS 

Kind regards,

Pamela Berkman

Publishing Editor, 

PLOS Biology

on behalf of

Gabriel Gasque,

Senior Editor

PLOS Biology